# Extracellular DNA as a Strategy to Manage Vascular Wilt Caused by *Fusarium oxysporum* in Tomato (*Solanum lycopersicum* L.) Based on Its Action as a Damage-Associated Molecular Pattern (DAMP) or Pathogen-Associated Molecular Pattern (PAMP)

**DOI:** 10.3390/plants13212999

**Published:** 2024-10-27

**Authors:** Alejandra Jiménez-Hernández, Ireri Alejandra Carbajal-Valenzuela, Irineo Torres-Pacheco, Enrique Rico-García, Rosalía V. Ocampo-Velazquez, Ana Angélica Feregrino-Pérez, Ramón Gerardo Guevara-Gonzalez

**Affiliations:** 1Center of Applied Research in Biosystems (CIAB-CARB), School of Engineering-Campus Amazcala, Autonomous University of Queretaro, Carr. Amazcala-Chichimequillas Km 1.0, S/N, El Marques C.P. 76926, QRO, Mexico; ale.jhtsu@gmail.com (A.J.-H.); carbajalireri@gmail.com (I.A.C.-V.); irineo.torres@uaq.mx (I.T.-P.); ricog@uaq.mx (E.R.-G.); rosov@yahoo.com.mx (R.V.O.-V.); 2Posgraduate Studies Division, C.A Basic and Applied Bioengineering, School of Engineering, Autonomous University of Queretaro, C.U Cerro de las Campanas, S/N, Colonia Las Campanas, Santiago de Querétaro C.P. 76010, QRO, Mexico; feregrino.angge@hotmail.com

**Keywords:** *Fusarium oxysporum*, eDNA, DAMP, PAMP

## Abstract

Vascular wilt is an important tomato disease that affects culture yields worldwide, with *Fusarium oxysporum* (*F.o*) being the causal agent of this infection. Several management strategies have lost effectiveness due to the ability of this pathogen to persist in soil and its progress in vascular tissues. However, nowadays, research has focused on understanding the plant defense mechanisms to cope with plant diseases. One recent and promising approach is the use of extracellular DNA (eDNA) based on the ability of plants to detect their self-eDNA as damage-associated molecular patterns (DAMPs) and pathogens’ (non-self) eDNA as pathogen-associated molecular patterns (PAMPs). The aim of this work was to evaluate the effect of the eDNA of *F.o* (as a DAMP for the fungus and a PAMP for tomato plants) applied on soil, and of tomato’s eDNA (as a DAMP of tomato plants) sprayed onto tomato plants, to cope with the disease. Our results suggested that applications of the eDNA of *F.o* (500 ng/µL) as a DAMP for this pathogen in soil offered an alternative for the management of the disease, displaying significantly lower disease severity levels in tomato, increasing the content of some phenylpropanoids, and positively regulating the expression of some defense genes. Thus, the eDNA of *F.o* applied in soil was shown to be an interesting strategy to be further evaluated as a new element within the integrated management of vascular wilt in tomato.

## 1. Introduction

In modern agriculture, one of the most important aspects is the management of diseases and pests, with vascular wilt being a determining disease in tomato [1,2], strongly related to the phytopathogenic fungi *Fusarium oxysporum* (*F.o*) and to high economic losses worldwide [3]. The management of this pathogen has been difficult due to its resilient spores being capable of persisting in the soil for more than 20 years and its capacity to reactivate when a possible host is detected [4]. The mechanism of infection of this fungus starts with the germinated conidia in soil and the growth of hyphae toward the plant roots, and its penetration is facilitated by the presence of wounds in plant tissues [3]. Once inside the tissues, mycelia keep growing, and if the pathogen reaches the xylem, obstruction of water and nutrients starts for specific plant parts and eventually the entire plant [5]. The symptoms of *Fusarium* wilt in tomato include yellowing of leaves, plant wilt and death. Also, a brown vascular coloration is observed in advanced infections [6]. Although in some cases *Fusarium* wilt symptoms are less severe, infected plants are commonly stunted and their yields are quite reduced. *Fusarium* species have a high capacity to evolve into different strains with diverse pathogenic mechanisms [7,8]. Thereby, the control of vascular wilt in tomato usually consists of the integration of multiple strategies, including cultural, physical and chemical control [6,9,10]. Each strategy provides benefits and disadvantages depending on the system where it is applied; thus, the selection of techniques is crucial to control the disease. To broaden the available techniques, researchers are continuously evaluating new management strategies. Another strategy to manage vascular wilt is biological control, which consists of the knowledge and management of natural defense mechanisms such as antagonistic ecological relations, microbial competition, growth promoting bacteria and plant immune system induction [1]. Current research has focused on the activation of the early defense pathways of the plant and the role of the elicitors in this natural process [11]. Elicitors are biological stress factors that trigger plant immunity in a dose–response manner [12,13,14]. Once the appropriate dose and application of a specific elicitor are determined, the induction of plant tolerance to biotic and abiotic stresses may be achieved. Some examples of applied elicitors with successful control of pests or diseases are methyl jasmonate [15], algae and organic compounds [16], pyrazines [17] and chitin [18].

Recently, researchers have reported the role of extracellular DNA (eDNA) as an elicitor when sensed outside the cell as a signal molecule in several organisms [19]. Moreover, eDNA released from the same organism (self-eDNA) has been evaluated as a “Damage-associated molecular pattern” (DAMP), which may induce innate immunity [20,21].

Evidence has shown that the plant immune system activation as a result of self-eDNA elicitation displays several responses in the plants. Among these responses are plasma membrane depolarization and Ca^2+^ cations accumulation [22], oxidative burst, activation of mitogen-activated protein kinases (MAPKs), induction of extra floral nectar [23], specific genetic differential expression [21,24,25] and the activation of the phenylpropanoids pathway and the activity of antioxidant enzymes [26]. Additionally, self-eDNA has been demonstrated to reduce the numbers of colony forming units (CFUs) per g of common bean after infection by the bacterium *P. syringae* [23].

However, in natural environments, the presence of eDNA is constant and is released from all the organisms living around, resulting in a mixture of self- and non-self eDNA acting as signal molecules. Specifically, DNA from microorganisms has been reported to elicit defense mechanisms in plants by acting as pathogen-associated molecular patterns (PAMPs). Some examples are the increase of H_2_O_2_ production and callose deposition in *A. thaliana* with applications of *E. coli* eDNA [27] and the increase of the total phenolics and flavonoids content and induced resistance to pathogenic infection in *C. annuum* with a pathogen DNA mixture application [28].

Lastly, self-eDNA, similar to other elicitors, has presented an autotoxic effect when presented in high doses [19,29,30]. This important issue has been proposed to be used as a new technology to keep the pathogenic communities controlled in an agricultural ecosystem (at high doses) while activating the plant immune system [20,31]. However, little research has been conducted to test eDNA treatment in its multiple approaches when facing a real pathogenic infection. Thus, the aim of this work was to evaluate the effect of the eDNA of *F.o* (as a DAMP for the fungus and a PAMP for tomato plants) applied on soil, and of tomato’s eDNA (as a DAMP of tomato plants) sprayed onto tomato plants, to cope with the disease.

## 2. Results

### 2.1. F. oxysporum Inhibition Assay in Soil Using Self-eDNA

Based on the known inhibitory effect of self-eDNA above-mentioned regarding non-self eDNA, in the present work, the self-eDNA of *F. oxysporum* (*F.o*) was evaluated in soil to determine the possible inhibitory responses in the fungus. According to Figure 1, in the first 24 h of inoculation, for the corrective treatment (CTS-500, using 500 ng/µL of self-eDNA in soil), where the application of eDNA was carried out after inoculation of the spores, a significant decrease was shown in the fungus levels with respect to the control, decreasing to one logarithmic unit at 120 h. As expected, the response of the fungus to the self-eDNA was dependent on the concentration, as observed in the results with the CTS-50 treatment (corrective treatment using 50 ng/µL of eDNA in soil) (Figure 1). In this case, the CFU levels were not different in comparison with the control during all the experiments. Interestingly, it was observed that although the same concentration of self-eDNA (500 ng/µL) was used in the preventive treatment (PTS-500), the CFUs were similar to those of the control (Figure 1). A possible explanation is that, in PTS-500, the eDNA was first applied within the soil and further inoculated with fungus spores; it is likely that the *F.o* spores sensed the DAMP presence (self-eDNA), thus inhibiting spore germination and growth in soil, but continued to be viable until cultured in the petri dishes, where they germinated at the same levels as the control. This latter asseveration is supported based on the percentage of spores reported in Figure 2B. In this figure, the fungus level from the soil samples that were pasteurized to eliminate vegetative cells can be observed, thus differentiating the fungal vegetative cells and the spore levels in the treatments [32]. During the first 72 h, the percentage of spores of the control and PTS-500 was around 62%, increasing at 168 h to around 67% (Figure 2B). The results shown in Figure 2A, using only the self-eDNA at 500 ng/µL, display that the development of the PTS-500 and control groups over time was similar (confirming the observation above). Moreover, at this time, the CTS-500 and CTS-2d (500 ng/µL eDNA applied 2 days after the fungus inoculation) treatments showed a significant decrease in the fungal levels at 72 h of inoculation and at 168 h (Figure 2A). Nevertheless, the percentage of spores increased during the first 72 h and was maintained at 87% during 72 to 168 h, being the treatment with more spores over the time due to the presence of its self-eDNA (Figure 2B).

### 2.2. Morphological Changes and Vascular Wilt Evaluation in Tomato Treated with eDNA of F. oxysporum and S. lycopersicum

The changes in the morphological variables in tomato plants caused by *F.o* infection in tomatoes treated or not with eDNA are shown in Figure 3. All the treatments evaluated, except the corrective one (CT), showed a similar response in comparison to the control plants (Figure 3). The corrective treatment displayed a significant decrease using the eDNA of *F.o* and *S.l* for the variables stem thickness and plant height, respectively (Figure 3B,C). The severity level of the disease is shown in Figure 4. It is clear in Figure 4B that the disease severity of the preventive or corrective treatments using *F.o* eDNA was significantly decreased compared with WPC, and not significantly different from the treatments of WC, WNC and C (treatments that had not been inoculated with the phytopathogen) (see Table 1). This latter result indicates that the disease severity observed in the corrective and preventive treatments using *F.o* eDNA responded similarly to the treatments in which no pathogen was inoculated, thus displaying plant protection based on this severity index.

As observed, for both eDNA sources, in the wounded positive control (WPC), where the wounded plants were exposed to only the fungus, the fourth level was reached according to the Popoola scale [33], being the treatment with the highest severity level. Interestingly, in the treatments with *S.l* eDNA as a plant DAMP, the corrective treatment (CT) showed a higher severity level than the wounded negative control, wounded control and control (WNC, WC and C) (Figure 4A); nevertheless, the preventive treatment (PT) showed a similar severity level to all the control treatments.

**Figure 4 plants-13-02999-f004:**
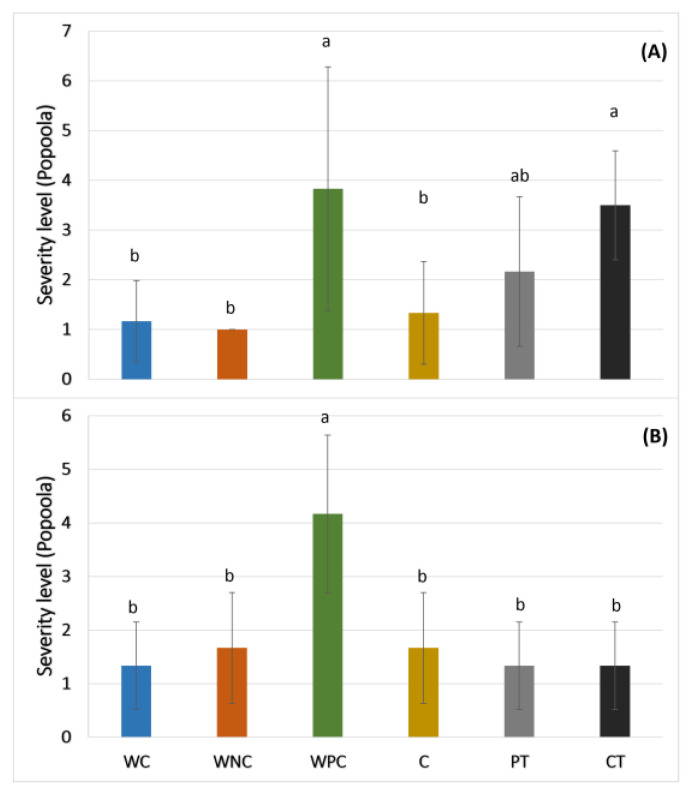
Severity level of the vascular wilt in tomato caused by *F.o* based on the scale reported by [34]. Tomato plants were treated with the eDNA of *S.l* (**A**), and *F.o* eDNA (**B**). WC = wounded control, WNC = wounded negative control, WPC = wounded positive control, C = control, PT = preventive treatment, CT = corrective treatment. Different lowercase letters in each group indicate significant differences (Kruskal–Wallis and Dunn’s test with α = 0.05).

### 2.3. Biochemical and Molecular Changes in Tomato Treated with eDNA of F. oxysporum and S. lycopersicum

Reactive oxygen species (ROS) are important molecules in signaling pathways such as growth, development and defense in plants [35]. Nevertheless, the overproduction of these molecules can generate oxidative injuries in the tissues, so the plants synthesize several enzymatic and non-enzymatic antioxidants like phenylpropanoids to cope with it [34]. Figure 5 shows the results of the total flavonoids and total phenolics in the tomato plants treated with eDNA from both sources. In the case of flavonoids, the plants showed no difference among the treatments in comparison to the control (Figure 5A). Other studies designed to cope with vascular wilt reported the induction of flavonoid levels using other types of elicitors for studying vascular wilt [35].

On the other hand, it is shown that the total phenolics significantly increased in the preventive and corrective treatments in the plants treated with *S.l* eDNA in comparison to the control (100% and almost 300% more than the control, respectively) (Figure 5B). However, the plants treated with *F.o* eDNA displayed a similar result in the corrective treatment (Figure 5B), increasing 100% more than the control, being similar to the level reached for the WPC group. In this case, the PT group showed the same production of phenolics as the control and wounded control (WC). These results are important when compared with the severity levels of disease in the preventive and corrective treatments, suggesting that the protection against vascular wilt activated by the eDNA treatments is related to the phenolic compounds in tomato, as reported before [35,36]. Additionally, it is clear that *F.o* infection per se significantly increased the total phenolics levels; however, the eDNA treatments (especially CT using *S.l* eDNA) seemed to increase the phenolics contents for plant defense. Maybe the types of specific phenolic compounds induced by eDNA are important to cope with vascular wilt in the model study evaluated in this work. Future studies should evaluate the contents of specific phenolics (and likely also flavonoids) and not only the total contents of them induced by eDNA treatments, trying to unravel in more detail the possible role of them in this plant–pathogen interaction.

The use of self-eDNA as a plant DAMP has recently been studied, showing not only the biochemical response of the plants inducing resistance against pathogens but also the elicitation of the expression of some defense genes [29,37]. In addition to the phenylpropanoids levels in the present work, the expression of some defense genes related to systemic acquired resistance (SAR) [38], *PR1a* (pathogenesis-related protein 1a) and *CHS* (chalcone synthase) was carried out in the preventive treatments as a model of study (Figure 6). On the one hand, a significant up-regulation of *PR1a* expression was observed in the plants treated with *F.o* eDNA (Figure 6A). On the other hand, the *CHS* gene expression in the tomato plants was significantly up-regulated with eDNA from both *S.l* and *F.o* (Figure 6B).

## 3. Discussion

In nature, all organisms coexist and develop sensitivity to danger signals, particularly danger caused by other organisms, in order to activate immune responses to survive [39,40]. The perception of the organisms is mainly due to the exogenous (PAMPs) and endogenous (DAMPs) danger molecular signals that trigger the immune system [41,42]. In this study, the use of *F.o* eDNA as a fungus DAMP as well as a tomato PAMP and *S.l* eDNA as a tomato DAMP was evaluated. The results showed that the *F.o* eDNA had an inhibitory effect on spore germination at 500 ng/µL since the first 24 h of the inoculation, as reported by Ferrusquía-Jiménez et al. with other phytopathogen (*Phytophthora capsici*) [29]. In this latter study, the authors obtained a suppressed zoospore germination of an oomycete at 500 ng/µL of self-eDNA doses. Furthermore, the application time was evaluated to study the best response, as shown in Figure 2B, where the percentage of spores was higher in the treatment CTS-2d (applied 2 days after inoculation). This result suggested that the application of *F.o* eDNA in soil once the spores have germinated is a possible management strategy to decrease the colonies of *F.o* in soil by inhibiting the vegetative tissue and preventing new spore germination. It should be noted that the application of eDNA in the treatments occurred only once during the experiment due to the knowledge that the eDNA from dead microorganisms can persist in soil for weeks to years, depending on the physical-chemical properties of the soil, being adsorbed by the soil particles such as clay, sand and other colloids [43,44]. Nevertheless, a study of the persistence of *F.o* eDNA under normal conditions in soils may be taken into account in future studies. Furthermore, the effect on the fungus may be different when the plant host is present [32]. In the absence of phytopathogens, the plant development is optimal, but during pathogen attack, the morphological, physiological, biochemical and molecular regulation is affected [45]. However, the plant response is activated once the danger is sensed [30]. In this study, the tomato seedlings that were inoculated with *F.o* and treated with self- and non-self eDNA showed morphological differences. The group with corrective treatment was the most affected in terms of the stem thickness in the plants that were treated with *F.o* eDNA, and in the case of the plants treated with *S.l* eDNA, the plant height was the most affected in the corrective treatment group. It is well known that once the immune responses are activated in plants, the energy consumption relocates, affecting their growth [23], as reported in [28], where a decrease was observed in the height and root density of infected chili plants treated with a mixture of the eDNA of the pathogenic complex. Furthermore, an evaluation of the severity levels according to the Popoola scale was performed, where the WPC group (wounded positive disease control) showed a level of 4 on the severity scale (Figure 4), which corresponded to moderate wilting with 11–20% of yellowed leaves and wilted in the cases of both the self- and non-self eDNA treatments. Additionally, in the WPC treatment, 33% of plants died. Interestingly, in the case of the *S.l* eDNA, the preventive (PT) and corrective treatments (CT) showed greater severity levels in comparison to the negative controls of disease (Figure 4A). These latter results suggest that at least in the case of *F.o* eDNA, the treated plants showed a higher immunity induction (total phenolics, *PR1a* and *CHS* up-regulation) than when using *S.l* eDNA, and this likely in part explains the protection against vascular wilt observed.

It is reported that the response of the elicited plants by DAMPs ranges from early signaling cascades inducing the innate immune system like the production of reactive oxygen species (ROS), elevated cytosolic Ca^2+^, and MAPK activation to phenotypic resistance responses by defense genes [20,29,42]. The defense of the plants is associated with structural features and the production of secondary metabolites [35] such as phenolics and flavonoids, which were evaluated in the tomato seedlings, showing an increase in phenolics following the PT and CT treatment (preventive and corrective treatment, respectively) in the case of *S.l* eDNA, and PT with *F.o* eDNA, in comparison to the control. Interestingly, the results obtained in terms of the molecular response, where the expressions of genes such as *PR1a* and *CHS* showed an up-regulation in the PT using *F.o* eDNA, showed both genes to be associated with the systemic acquired response [26]. It is noteworthy that the differential gene expression was evaluated at the end of the experiment, suggesting that the activation of the immune system by *F.o* eDNA in tomato plants when *F.o* is infecting the plant remains active up to 4 weeks. As mentioned above, this continuous energy spending may be translated into lower growth or stem thickening rates in older stages. In contrast, the control plants showed a down-regulated gene expression. Perhaps a second application of the treatment for the maintenance of the activated state should be evaluated in future studies. It would be important to evaluate in further studies the dynamic of gene expression and even more SAR gene markers in order to unravel in more detail this phenomenon. Although, in general, the protection obtained with both sources of eDNA to cope with *F.o* infection was adequate, it is important to notice the difference between the applied dose and the application mode. While *S.l* eDNA was applied at 50 ng/µL directly in the foliage of tomato seedlings, *F.o* eDNA was applied at 500 ng/µL in the soil in order to obtain differences in the immunity and plant protection. This evidence displayed, as expected, that plants are more sensitive to self-eDNA (*S.l* eDNA). Even though the dose was lower than the application of non-self eDNA (*F.o* eDNA), this may be important in order to evaluate the viability of treatments in an agricultural system. Moreover, agricultural variables such as the yield and fruit quality in an infected culture may be taken into account in disease management strategies. Some disease management goals are not to totally cut out the infection but to decrease the symptoms to a level such that the production will not be diminished while maintaining a more equilibrated agroecosystem [46]. Modern management focuses on the immune system level of the plant and the environment, and not only in the presence of a certain pathogen (according to the triangle of the disease) [47]. Also, we would like to highlight that eDNA treatments in agriculture are highly sustainable, with the organic residues of the culture or microorganism cultivation being excellent raw materials to obtain eDNA, as detailed in [48].

## 4. Materials and Methods

### 4.1. Biological Material

Tomato seeds of the variety RIO GRANDE from King Seeds Inc. (Guadalajara, México) were used in this study. In this study, the use of this cultivar was chosen based on the fact that producers in Central Mexico use this variety because of its commercial importance, it is an economically available variety, and it is susceptible to the *Fusarium oxysporum* strain used in this work. The strain of *F. oxysporum* was kindly provided by the Instituto Nacional de Investigaciones Forestales, Agrícolas y Pecuarias (INIFAP) campo experimental Bajío. The pathogen was isolated from infected tomato culture soil in Central Mexico (Silao, México) and was morphologically and molecularly identified using the specific primers for *Fusarium oxysporum* reported in [49]. The pathogenicity of this strain was confirmed as strong in tomato plants (inoculated at the 4–6 true leaves stage) and typical vascular wilt symptomatology was observed at 7–10 days post-inoculation. This strain was also tested in chili pepper (4–6 leaves stage) and chickpea (4–6 leaves stage) plants and maintained under observation within a greenhouse for 4 weeks, but no symptoms were detected, thus determining that the *F. oxysporum* used in the present work is host-specific for tomato. Strain activation was carried out by the inoculation of tomato plantlets and further re-isolation and identification of the fungus from the plantlet tissues by surface sterilization and microbiological culture in a tomato leaf agar.

### 4.2. DNA Extraction

Tomato DNA extraction was performed from the pruned leaves of previously established tomato plants (3 months old). Briefly, 100 mL of extraction buffer (200 mM Tris-HCl pH 8.5; 250 mM NaCl; 25 mM EDTA; 0.5% (*w*/*v*) SDS) was mixed with 50 g of tomato leaves in a common kitchen blender Oster 6662-13 (Oster México, Acuña, México) for 10 s. The obtained mixture was separated with a metal sieve to discard the solid waste. Then, 15 mL of 3 M sodium acetate were added to 25 mL of the obtained liquid and mixed with a vortex. This mixture was rested for 30 min at 4 °C and then separated in a DM0412S (Science Med, Helsinki, Finland) centrifuge for 15 min at 8500 rpm and 4 °C. Next, 25 mL of the obtained supernatant was mixed gently with 15 mL of cold isopropanol. The tubes were rested at −20 °C for 30 min and centrifuged again under the same conditions mentioned above. At this time, the supernatant was discarded and the obtained pellet was dried in the open tubes at room temperature for 24–48 h. When fully dry, the pellet was resuspended in distilled water.

The fungal DNA extraction was performed according to Dellaporta et al. [50], where 2 g of *F.o* mycelium was weighed into 50 mL falcon tubes previously crushed in liquid nitrogen, and then 10 mL of Dellaporta buffer (100 mM Tris-HCl, 520 mM EDTA, 2% PVP, 500 mM NaCl, 10 mM β-mercaptoethanol) was subsequently added and shaken to homogenize the mixture. Once homogenized, 500 µL of 20% SDS was added and mixed carefully, then incubated at 65 °C for 20 min. Next, 3 mL of 5 M potassium acetate was added and placed on ice for 20 min, centrifuged at 8500 rpm for 25 min, and the supernatant was recovered in another tube. A volume of cold isopropanol was added to this new tube with the supernatant and left on ice for 10 min. Finally, it was centrifuged at 8500 rpm for 25 min and the supernatant was discarded. The obtained pellet was washed with cold 70% ethanol, allowed to air-dry and suspended in 5 mL of sterile distilled water, and the tubes were stored at −70 °C.

The DNA concentrations were measured in a NanoDrop 2000c spectrophotometer (Thermo Scientific, Logan, UT, USA) and the integrity visualized through an agarose gel electrophoresis in order to confirm the size of the DNA fragments. If the DNA fragment size was larger than 1 Kb, the solutions were sonicated with a UP200Ht sonotrode (Hielscher, Teltow, Germany) with pulses of 26 KHz at 10 W and 50% of amplitude every second for 30 min.

### 4.3. Biological Assays

#### 4.3.1. Fungus Inhibition Assay in Soil (Self-eDNA as a DAMP)

The soil used was analyzed for its texture and organic matter by the Walkley–Black method [51]. The results were as follows: loam soil with medium texture and 1.54% of organic matter. The experiment was carried out according to Figueroa-Rivera et al. (2010) [52], where 500 mg of soil was weighted into amber glass vials, sterilized at 220 KPa for 15 min, then inoculated with 2 × 10^9^ spores/mL of *F.o* at field capacity with distilled water. One hour later, 50 and 500 ng/µL concentrations of *F.o* eDNA (self-eDNA) were applied, which corresponded to 2500 and 25,000 ng of *F.o* eDNA, respectively, as shown in Table 1. These doses were defined according to previous results obtained with *Phytophthora capsici*, where these concentrations were able to suppress the zoospore germination under in vitro conditions [29]. At 0, 24, 72, 120 and 168 h after the inoculation, the soil was serially diluted in saline solution and cultivated in PDA plates. The plates were incubated for 48 h at 25 °C for their subsequent colony forming units (CFUs) quantification.

This assay allowed us to know the best concentration of *F.o* eDNA to reduce the colony forming units (CFUs) in soil. Based on these results, another assay like the previous one was established using only the concentration of 500 ng/µL of *F.o* eDNA and adding a new group, which consisted of the application of the spores and two days later the *F.o* eDNA (CTS-2d), to verify what was observed. To determine the best way to apply the *F.o* eDNA in soil, the other treatments (CTS-500 and PTS-500) were carried out. Furthermore, a group of each treatment was exposed to a pasteurization consisting of a water bath at 70 °C for 1.5 h [53,54] 1 hour before cultivating each petri dish. This was performed knowing that vegetative cells are eliminated above 50 °C [32], unlike the spores, which remain viable under these conditions. This way, the spores that germinated through the experiment were now vegetative tissue that was eliminated with pasteurization. Thus, the obtained CFUs from the pasteurized group were germinating from spores and not from vegetative cells. From these data, we can determine the percentages of spores in each treatment CFU (Figure 2B).

#### 4.3.2. Disease Control with *F.o* eDNA (Self-eDNA as Pathogen DAMP and Plant PAMP) and with *S.l* eDNA (Self-eDNA as Plant DAMP)

Tomato seedlings were tested once they had 2 true leaves, where 6 plants were used for each treatment. The tomato seedlings were transplanted to a clean and disinfected tray using previously sterilized soil. For some treatments, a slight wound to the lower third of the roots was made in the seedlings to facilitate the infection, and they were soaked in fungal inoculum at 1.92 × 10^9^ spores/mL. This was performed for the wounded positive disease control (WPC) group as well as the preventive and corrective treatments (PT and CT). In the case of the other control groups (wounded control WC and wounded negative control WNC), the slight wound to the root was made and instead of being soaked in fungal inoculum, these groups were soaked in either distilled water for WC or *F.o* eDNA was applied for WNC. Furthermore, a control treatment (C) consisted of plants that were neither wounded nor soaked in distilled water was carried out. Once transplanted, the treatments were added to the soil according to Table 2 and leaving the soil at field capacity. It should be noted that the doses applied in the soil of *F.o* eDNA were the 500 ng/µL in the WNC, PT and CT groups, which corresponded to 1,000,000 ng of *F.o* eDNA. Finally, the tomato seedlings were monitored for 28 days.

At the same time, another assay was carried out, just like the one described above, to compare the effects of DNA as a DAMP or a PAMP in the management of the disease in tomatoes. In this case, the self-eDNA of tomato (*S.l* eDNA) was used instead of the eDNA of the fungus. A concentration of 1.92 × 10^9^ spores/mL fungal inoculum was applied in sterilized soil and *S.l* eDNA at 50 ng/µL was sprayed onto the foliage of tomato seedlings at a drop point using approximately 3 mL per plant, which corresponded to 150,000 ng of *S.l* eDNA in only the treatments specified in Table 2 (WNC, PT and CT).

### 4.4. Growth Variables

From the start of the study, the physiological state of the seedlings was evaluated twice a week by measuring the plant height with a plastic ruler and the stem thickness with a vernier caliper.

### 4.5. Severity Level Assessment

The disease index scoring the severity of the plant–fungus assays was classified according to the disease index scale of Popoola et al. (2015) [33]. In this scale, the scores used are the following: 1 = stems and leaves free of any visual symptoms, 2 = very limited wilting, 5% leaves yellowed and wilted, 3 = limited wilting, 6–10% leaves yellowed and wilted, 4 = moderate wilting, 11–20% leaves yellowed and wilted, 5 = severe wilting, 21–50% leaves yellowed and wilted, 6 = very severe wilting, above 50% leaves yellowed and wilted. Once the tomato seedlings were monitored for 28 days, we removed each plant and washed the roots with distilled water, and observations were made of the root system and foliage to determine on which scale level it should be classified.

### 4.6. Total Phenolics and Flavonoids Determination

Methanolic extracts of the samples were obtained by crushing the vegetal tissues with liquid nitrogen and lyophilized. Then, 200 mg was weighed and 10 mL of methanol was added. The mixture was sonicated for 30 min and centrifuged at 5000 rpm and 4 °C for 10 min. The supernatant was recovered and refrigerated until further analysis. The total phenols were determined by mixing 460 µL of distilled water and 40 µL of methanolic extract with 250 µL of Folin–Ciocalteu phenol reagent (Sigma-Aldrich, Saint Lous, MO, USA), left to rest for 5 min and then 1250 µL of 20% Na_2_CO_3_ was added and the mix was left to rest for 2 h at room temperature before reading its absorbance at 760 nm, all of this in a low light environment [55,56]. Flavonoids were determined from the same methanolic extract by mixing 180 µL of distilled water and 50 µL of methanolic extract with 20 µL of 1% 2-aminoethyl diphenylborinate methanolic solution and measuring the absorbance at 404 nm [57]. Measurements were obtained with a Multiskan Sky Hich (Thermo Scientific, Logan, UT, USA) spectrophotometer and calibration curves were obtained using gallic acid for phenols and quercetin for flavonoids. The values are reported as a percentage of total phenolics and flavonoids with respect to the control. These percentages were obtained once the total phenolics were reported as mg of gallic acid equivalents/mg of dry tissue, and the total flavonoids as mg of quercetin equivalents/mg of dry tissue, while the value of the control group in both polyphenolics was taken as 100%.

### 4.7. Gene Expression Analysis

Leaf samples were homogenized with liquid nitrogen until a fine powder was obtained and stored at −40 °C until analysis. The total RNA of each sample was isolated based on the TRIzol method reported in [38]. Tissue powder was mixed with 1 mL of TRIzol™ (Thermo Fisher, Carlsbad, CA, USA) and incubated at room temperature for 10 min. Then, 600 µL of chloroform was added, the mixture was vortexed for 1 min and centrifuged at 12,000 rpm and 4 °C for 15 min. Once separated, the upper phase was collected in a new tube and mixed with 600 µL of cold isopropanol. The mix was incubated for 1 h at −40 °C and then centrifuged again. The supernatant was discarded and the obtained pellet was washed with 200 µL of 75% ethanol. The washed pellet was air-dried at room temperature for 20–30 min and then resuspended in 50 µL of sterile distilled water.

The *PR1a* and chalcone synthase (*CHS*) gene expression was determined by reverse transcription (RT)-qPCR as follows: the obtained RNA was evaluated by agarose gel electrophoresis and quantified with a NanoDrop™ 2000 spectrophotometer from Thermo Fisher. The RNA concentration of all the samples was equalized and cDNA was synthesized from it using the Maxima First Strand cDNA Synthesis Kit from Thermo Fisher following the manufacturer’s indications. This step was followed by qPCR with the Maxima SYBR Green/ROX qPCR Master Mix (2x) from Thermo Scientific™ and the following primers: forward: 5′-GCCAAGCTATAACTACGCTACCAAC-3′ and reverse: 5′-GCAAGAAATGAACCACCATCC-3′ for PR1a; forward: 5′-CCAAGGACTTGGCTGAGAAC-3′ and reverse: 5′-TATCGGGGACAAGAGTTTGG-3′ for chalcone synthase and forward: 5′-AGTTGCCCCAGAAGAACACC-3′ and reverse: 5′-CCACCACCTTGATCTTCATG-3′ for actin, as reported before [38]. The PCR program was as follows: 95 °C for 10 min for initial denaturation, then 40 cycles of denaturation, annealing and extension (95 °C 15 s, 55 °C 30 s and 72 °C 30 s), and finally, an extension step of 72 °C for 30 s in a real-time system from BIORAD Laboratories. The relative gene expression levels were calculated using the ∆∆Ct method using actin as the housekeeping gene.

### 4.8. Statistical Analysis

Physiological, biochemical and molecular variables from the evaluated data were analyzed by a one-way ANOVA (alpha = 0.05) and a Tukey test. For the severity index, the hypothesis were tested by a Kruskal–Wallis and Dunn test with alpha = 0.05. 

## 5. Conclusions

The use of the self-eDNA of *F.o* applied in soil as a preventive or corrective treatment showed protection against tomato vascular wilt caused by *F.o*. The latter because it significantly decreased the severity levels and improved the plantlets’ development. This phenotype was associated with the biochemical and molecular responses due to the increase in the production of total phenolics, as well as the expression of some genes related to systemic acquired resistance (SAR). Furthermore, the use of *F.o* self-eDNA (as a DAMP for the fungus and a PAMP for the tomato plants) in soil might be a possible treatment in the future to clean up soils contaminated with this pathogen and for vascular disease management.

## Figures and Tables

**Figure 1 plants-13-02999-f001:**
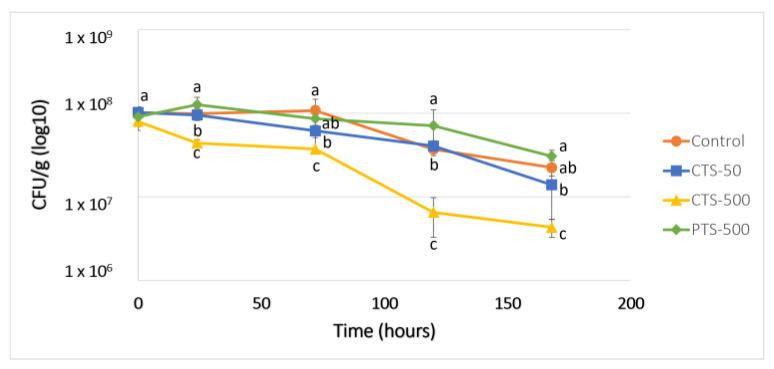
*F.o* CFUs in soil treated with different concentrations of *F. o* eDNA. CTS-50 = corrective treatment in soil at 50 ng/µL eDNA, CTS-500 = corrective treatment in soil at 500 ng/µL eDNA, PTS-500 = preventive treatment in soil at 500 ng/µL eDNA. Different lowercase letters indicate significant differences (two-way ANOVA and Tukey’s test with α = 0.05).

**Figure 2 plants-13-02999-f002:**
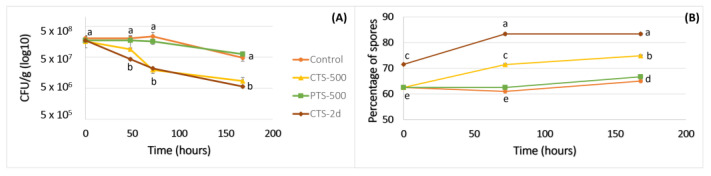
Panel (**A**) *F.o* CFUs in soil treated with *F o* eDNA at 500 ng/µL in different orders of application respecting the spore inoculation, and panel (**B**) percentage of spores during the time. CTS-500 = corrective treatment in soil at 500 ng/µL eDNA, PTS-500 = preventive treatment in soil at 500 ng/µL eDNA, CTS-2d = corrective treatment in soil at 500 ng/µL eDNA applied 2 days after the fungus inoculation. Different lowercase letters indicate significant differences (two-way ANOVA and Tukey’s test with α = 0.05).

**Figure 3 plants-13-02999-f003:**
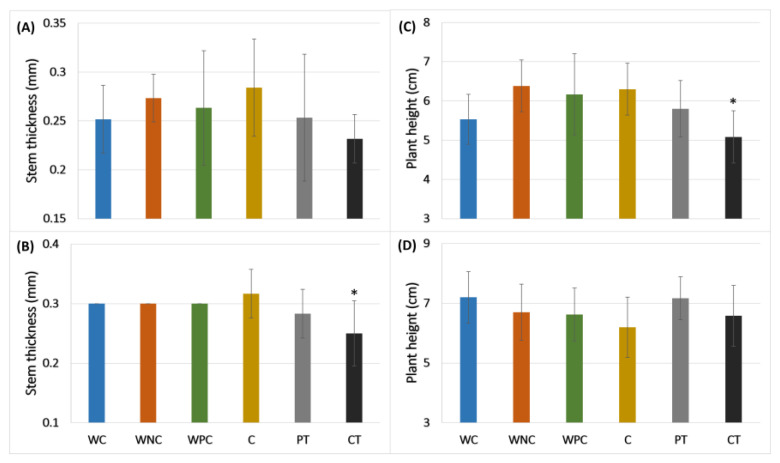
Morphological changes in tomato plants treated with two sources of eDNA (*F.o*, or *S.l*) and inoculated with spores of *F.o*. Panels (**A**,**B**), stem thickness of plants treated with *S.l* eDNA (**A**) and *F.o* eDNA (**B**). Panels (**C**,**D**), plant height in treatments with *S.l* eDNA (**C**) and *F.o* eDNA (**D**). WC = wounded control, WNC = wounded negative control, WPC = wounded positive control, C = control, PT = preventive treatment, CT = corrective treatment. * indicates statistical difference with the control (two-way ANOVA and Tukey’s test with α = 0.05).

**Figure 5 plants-13-02999-f005:**
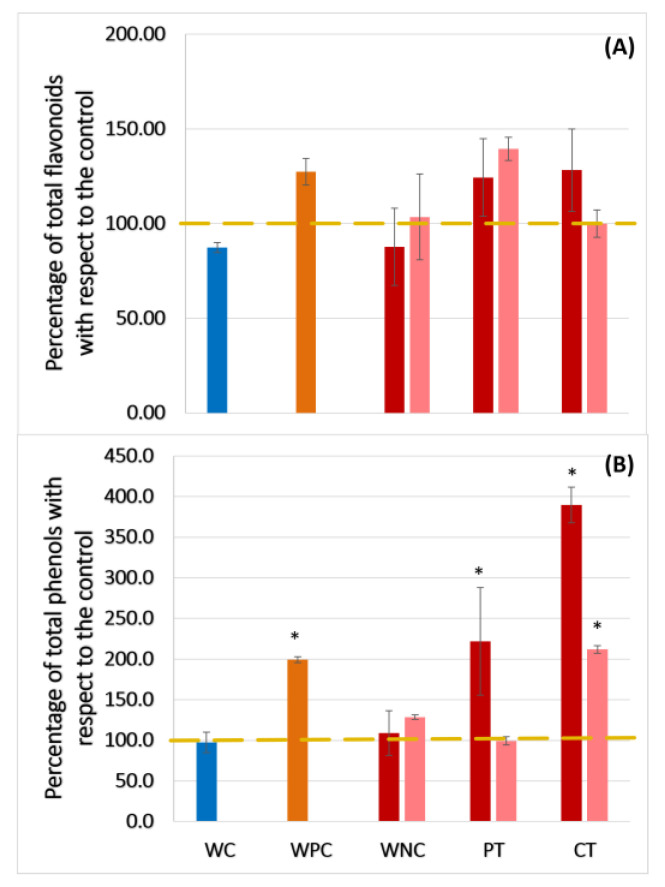
Total flavonoids determination in plants (Panel (**A**)). Total phenols determination in plants (Panel (**B**)). WC = wounded control, WPC = wounded positive control, WNC = wounded negative control, PT = preventive treatment, CT = corrective treatment. The dotted line represents the control (no wounded, not *F.o* inoculated and not treated with eDNA). The red bars represent the plants treated with *S.l* eDNA, and the rose bars represent the plants treated with *F.o* eDNA. * indicates statistical difference with the control (one-way ANOVA and Tukey’s mean test with α = 0.05).

**Figure 6 plants-13-02999-f006:**
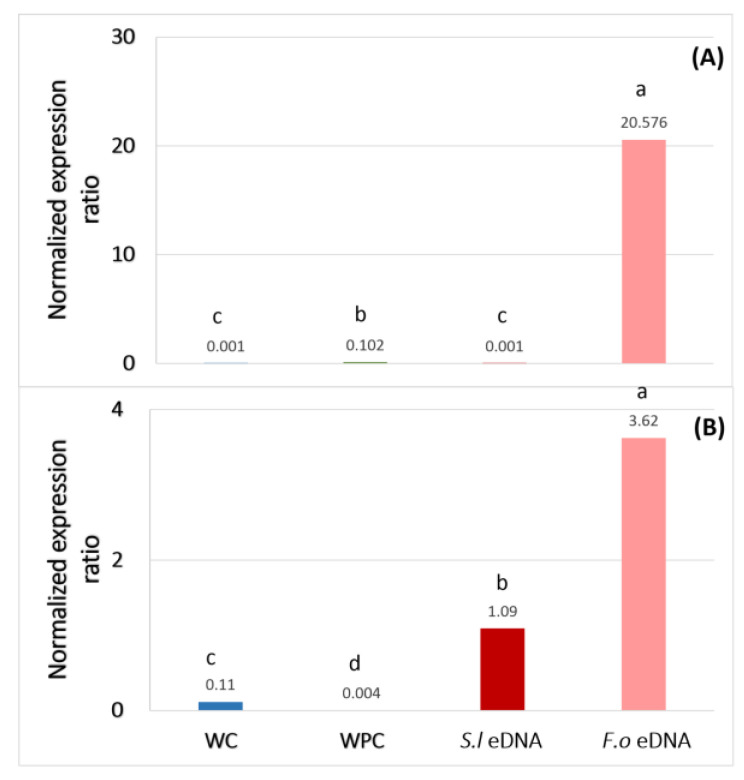
Gene expression of *PR1a* (**A**) and *CHS* (**B**). WC = wounded control, WPC = wounded positive control, *S.l* eDNA = preventive treatment with tomato eDNA, *F.o* eDNA = preventive treatment with *F.o* eDNA in tomato. Different lowercase letters in each group indicate significant differences (one-way ANOVA and Tukey’s mean test with α = 0.05).

**Table 1 plants-13-02999-t001:** Treatments carried out to determine the effect of *F.o* eDNA in its survival in soil.

Treatment	Description and Order of Application
Control	500 mg soil + 100 µL of fungus spores + 50 µL distilled water
CTS-50	500 mg soil + 100 µL of fungus spores + 50 µL eDNA at 50 ng/µL
CTS-500	500 mg soil + 100 µL of fungus spores + 50 µL eDNA at 500 ng/µL
PTS-500	500 mg soil + 50 µL eDNA at 500 ng/µL + 100 µL of fungus spores

Note: CTS = corrective treatment in soil, PTS = preventive treatment in soil.

**Table 2 plants-13-02999-t002:** Treatments of infection control with *F.o* eDNA.

Groups	Wounded Root	Description and Order of Application
Wounded control (WC)	X	20 g soil + 6 mL sterile distilled water
Control (C)	-	
Wounded positive control (WPC)	X	20 g soil + 6 mL of fungus spores
Wounded negative control (WNC)	X	20 g soil + 2 mL eDNA at 500 ng/µL + 4 mL sterile distilled water
Preventive treatment (PT)	X	20 g soil + 2 mL eDNA at 500 ng/µL + 4 mL fungus spores
Corrective treatment (CT)	X	20 g soil + 4 mL fungus spores and 2 days later 2 mL eDNA at 500 ng/µL

## Data Availability

The data generated during this study are available from the corresponding author on reasonable request.

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
