# Peer review of "Extracellular DNA as a Strategy to Manage Vascular Wilt Caused by *Fusarium oxysporum* in Tomato (*Solanum lycopersicum* L.) Based on Its Action as a Damage-Associated Molecular Pattern (DAMP) or Pathogen-Associated Molecular Pattern (PAMP)"

_plants, 2024, doi:10.3390/plants13212999_

Round 1

Reviewer 1 Report (Previous Reviewer 1)

Comments and Suggestions for Authors

Dear authors,

I read your answer for my comments and questions.

Your answer is so honest. Then I read your resubmitted manuscript.

Introduction → OK

Materials and Methods OK

Results

Figure 5 shows tomato plants that were treated with S. l. eDNA indicating in Fig 4 A. Therefore these tomato photos cannot show the significant effects of eDNA.

If you have the photos about treated with F. o. eDNA, the significant differences between WPC and eDNA treatment plots are more clearly.

Discussion OK

References [56] is not from journal article or book chapter. It’s like note.

Therefore it should be replaced.

Then your manuscript would be accepted, I think.

Best regards,

Author Response

Reviewer 1

Dear authors,

I read your answer for my comments and questions.

Your answer is so honest. Then I read your resubmitted manuscript.

Introduction → OK

Materials and Methods → OK

Address: Many thanks for your kind comments.

Results

Figure 5 shows tomato plants that were treated with S. l. eDNA indicating in Fig 4 A. Therefore these tomato photos cannot show the significant effects of eDNA.

If you have the photos about treated with F. o. eDNA, the significant differences between WPC and eDNA treatment plots are more clearly. Discussion → OK

Address: Again many thanks and agree. Considering these latter 2 comments and based on the opinion of other reviewer, we choose to delete Figure 5 because really it is not informative. Additionally, it would be great to get the photos you request to improve the Figure but it is not possible in this moment because unfortunaley, we did not took pictures of that treatment.

References → [56] is not from journal article or book chapter. It’s like note. Therefore it should be replaced

Address: Thanks and agree again. We have corrected this reference in the revised manuscript.

Then your manuscript would be accepted, I think.

Address: Again many thanks for all your kind comments.

Best regards,

Reviewer 2 Report (Previous Reviewer 2)

Comments and Suggestions for Authors

The authors declaimed that application of eDNA of F. o (500 ppm) in soil significant lowered disease severity levels in tomato, increasing the content of some phenylpropanoids, and positively regulating the expression of defense genes. However, there are still some issues needed to be solved in this manuscript.

1. In line 152-153, The authors stated that “It is clear that only preventive or corrective treatments using F.o eDNA significantly decreased disease severity (Figure 4A and B).”. Actually, from Figure 4, disease severity of preventive or corrective treatments using F.o eDNA was only significantly decreased than WPC, and not significantly differed from treatments of WC, WNC and C. The authors should discuss why this is.

2. In addition, the authors should give more detailed information to distinguish the treatments of WC= wounded control, WNC= wounded negative control, WPC= wounded positive control, C= control, PT= preventive treatment and CT= corrective treatment.

3. In Figure 1, the letter "a" is not marked at time 0, which should be added.

4. In Figure 7, the error bars were not marked. In figure legend, the names of genes should be uppercase and italicized as “PR1” and “CHS”. Please check the full manuscript for the similar formatting issues.

Author Response

Reviewer 2

The authors declaimed that application of eDNA of F. o (500 ppm) in soil significant lowered disease severity levels in tomato, increasing the content of some phenylpropanoids, and positively regulating the expression of defense genes. However, there are still some issues needed to be solved in this manuscript.

  1. In line 152-153, The authors stated that “It is clear that only preventive or corrective treatments using F.oeDNA significantly decreased disease severity (Figure 4A and B).”. Actually, from Figure 4, disease severity of preventive or corrective treatments using F.oeDNA was only significantly decreased than WPC, and not significantly differed from treatments of WC, WNC and C. The authors should discuss why this is.

Address: Many thanks for your kind comment. We agree with the comment and corrected (highlighted in green) in the revised version.

  1. In addition, the authors should give more detailed information to distinguish the treatments of WC= wounded control, WNC= wounded negative control, WPC= wounded positive control, C= control, PT= preventive treatment and CT= corrective treatment.

Address: We agree again and now this aspect is corrected in the revised version in the section of methods (highlighted in green).

  1. In Figure 1, the letter "a" is not marked at time 0, which should be added.

Address: Thanks and corrected in the revised version

  1. In Figure 7, the error bars were not marked. In figure legend, the names of genes should be uppercase and italicized as “PR1” and “CHS”. Please check the full manuscript for the similar formatting issues.

Address: Thanks again. This aspect has been corrected in the new Revised version. Now this comment is located in the Figure 6 (which in the former version it was Figure 7), because based on the comments of other 2 reviewers we deleted the originally numbered Figure 5 in the former revised version.

Reviewer 3 Report (Previous Reviewer 3)

Comments and Suggestions for Authors

The new version of manscript is much better than before, it almost can be accepted now. But I suggest to delete the figure 5, because not much useful information is contained. 

  minor concern: line 153, Figure 4B but no A

Author Response

Reviewer 3

The new version of manuscript is much better than before, it almost can be accepted now. But I suggest to delete the figure 5, because not much useful information is contained. 

Address: We agree with your comment. As you suggested we deleted the Figure 5 in this new revised version of the manuscript. Many thanks.

minor concern: line 153, Figure 4B but no A

Address: Thanks and corrected in the revised version (highlighted in gray)

Reviewer 4 Report (New Reviewer)

Comments and Suggestions for Authors

1. Please provide identification details, including the accession number for the pathogen (Fusarium oxysporum).

2. Is the pathogen host-specific?

3. Please cite the reference details for the Walkeley-Black method.

4. Kindly convert the DNA concentration from ppm to nanograms.

5. Authors are requested to clarify the methods used to assess disease severity and disease index.

6. The possibility of eDNA degradation in soil has been overlooked here. Please clarify. If it is not degraded, how much was recovered?

7. The author conducted qRT-PCR on the pr1a and chalcone synthase (CHS) genes for expression analysis. Why is this relevant for understanding PAMP/DAMP-mediated defense? I suggest studying the expression of pattern recognition receptor genes associated with eDNA. Additionally, the in-planta concentration of Fusarium oxysporum should be quantified to substantiate the claim of eDNA as a PAMP/DAMP.

Author Response

Reviewer 4

  1. Please provide identification details, including the accession number for the pathogen (Fusarium oxysporum).

Address: Many thanks for your comment. In the revised version 2 it is mentioned in more detail the information of identification of the F. oxysporum used in the study. The Institute that provided us the strain does not have or report any accession number. Only we have the information we now mentioned in this revised version (highlighted in blue). We also included the reference 51, used to molecular identification of F. oxysporum using specific primers.

  1. Is the pathogen host-specific?.

Address: The information of the Institute that provided us the strain indicated that this strain is specific to tomatoto (they informed us that also tested in chili and chickpea but no symptoms were detected). This aspect is mentioned highlighted in blue in the section of methods. Thanks again for this comment.

  1. Please cite the reference details for the Walkeley-Black method.

Address: Thanks for the comment again. This reference is now included in the revised version (Reference 53).

  1. Kindly convert the DNA concentration from ppm to nanograms.

Address: Thanks and done in the methods section. In this section we now mention the equivalent of the eDNA concentrations used (in ng/mL and the corresponded nanograms applied in each treatment). We omitted the use of ppm in the revised version.

  1. Authors are requested to clarify the methods used to assess disease severity and disease index.

Address: Thanks again for your comment. In this revised 2 version, we include a more detailed explanation on how we assessed the disease severity scale in the study for our evaluated treatments, hoping is OK for you. (highlighted in blue).

  1. The possibility of eDNA degradation in soil has been overlooked here. Please clarify. If it is not degraded, how much was recovered?

Addreess: Thanks and agree. In fact this is an excellent and important comment. We did not evaluate this aspect in this study, however in future studies in our group this aspect is considered to be evaluated. Some discussion regarding this comment is now included in the revised version (See Disscusion section, highlighted in blue).

  1. The author conducted qRT-PCR on the pr1a and chalcone synthase (CHS) genes for expression analysis. Why is this relevant for understanding PAMP/DAMP-mediated defense? I suggest studying the expression of pattern recognition receptor genes associated with eDNA. Additionally, the in-planta concentration of Fusarium oxysporum should be quantified to substantiate the claim of eDNA as a PAMP/DAMP.

Address: Thanks again for your kind comment. However, in the present study we only evaluated the SAR e ISR gene markers based on the following aspects, however we appreciate your comment and we are convincing that this type of studies you mention will enrich future knowledge in the eDNA field in plant science.  Earlier studies have been made focusing on elucidate the process in which the plants recognize DNA as a PAMP or a DAMP but although self-eDNA has shown to remain outside the plant cell indicating that plant cells are able to perceive self-eDNA as a DAMP at the cell surface (Zhou et al., 2023). No specific receptor (PRRs) have been identified in plants so far, unlike in mammals were several protein receptors are already known to recognize extracellular DNA. To detect the effect of DNA as a signal molecule (DAMP or PAMP) the immune system have been monitored by our group and others in several ways such as ROS metabolism and activation of the phenylpropanoids biosynthetic way, resulting in the synthesis of defense molecules such as phenolic substances and flavonoids (CarbajalValenzuela et al., 2022; Duran-Flores & Heil, 2015; Ferrusquía-Jiménez et al., 2022). With this evidence, we suggested PR1a and CHS as simple markers of the plant immune system activation but we agree that exploring other genes may be interesting to characterize the DAMP/PAMP effect more finely. On the other side, our group suggest that the elicitation of immune system may protect the plant in front of microbial infections by decreasing the effects that the microbe has in the plant but not by eliminate the pathogen. We thought that the overall result (plant infection severity) may be more conclusive for measuring the protection that these treatments activate in plants. Also we agree that the quantification of Fusarium inside the plant could be in help to understand the process. Below are some references of our group and others regarding the aspects abovementioned in the address of this comment. Many thanks in fact for this comment again.

 Carbajal-Valenzuela, I. A., Guzmán-Cruz, R., González-Chavira, M. M., Medina-Ramos, G., SerranoJamaica, L. M., Torres-Pacheco, I., Vázquez, L., Feregrino-Pérez, A. A., Rico-García, E., & Guevara-González, R. G. (2022). Response of Plant Immunity Markers to Early and Late Application of Extracellular DNA from Different Sources in Tomato (Solanum lycopersicum). Agriculture, 12(10), 1587. https://doi.org/10.3390/agriculture12101587

Duran-Flores, D., & Heil, M. (2015). Growth inhibition by self-DNA: A phenomenon and its multiple explanations. New Phytologist, 207(3), 482–485. https://doi.org/10.1111/nph.13542

Ferrusquía-Jiménez, N. I., Serrano-Jamaica, L. M., Martínez-Camacho, J. E., Sáenz de la O, D., Villagomez-Aranda, A. L., González-Chavira, M. M., Guerrero-Aguilar, B. Z., Torres-Pacheco, I., Feregrino-Pérez, A. A., Medina-Ramos, G., & Guevara-González, R. G. (2022). Extracellular self-DNA plays a role as a damage-associated molecular pattern (DAMP) delaying zoospore germination rate and inducing stress-related responses in Phytophthora capsici. Plant Pathology, 71(5), 1066–1075. https://doi.org/10.1111/ppa.13545

Zhou, X., Gao, H., Zhang, X., Rahman, M. K. u, Mazzoleni, S., Du, M., & Wu, F. (2023). Plant extracellular self-DNA inhibits growth and indunces immunity via the jasmonate signaling pathway. Plant Phyisiology

Round 2

Reviewer 2 Report (Previous Reviewer 2)

Comments and Suggestions for Authors

As most of the issues concerned were solved, I suggest that the manuscript might be accepted.

Author Response

Comments and Suggestions for Authors

As most of the issues concerned were solved, I suggest that the manuscript might be accepted.

Address:

Thanks for your comments.

Reviewer 4 Report (New Reviewer)

Comments and Suggestions for Authors

Dear Authors,

1. If you do not have an accession number, please include the location and name of the county.

2. Include details of chili and chickpea, but no symptoms were detected in the discussion section.

Author Response

Reviewer 4 (Round 2)

Comments:

Dear Authors,

  1. If you do not have an accession number, please include the location and name of the county.
  2. Include details of chili and chickpea, but no symptoms were detected in the discussion section.

Address:

Thank you again for your new comments. Now in the revised version 3 of this submission, the address to your 2 comments are located in the methods section and both are highlighted in blue.

Thanks.

This manuscript is a resubmission of an earlier submission. The following is a list of the peer review reports and author responses from that submission.

Round 1

Reviewer 1 Report

Comments and Suggestions for Authors

Dear correspondence author; Prof. Dr. Guevara-Gonzales,

 Professor’s Group already has been published the article about tomato disease in “Plants” in 2022.

Then you are excellent research group, I know.

But regarding to this manuscript is not excellent.

1 Experimental design is so poor.

 The aim is good, but Materials and methods are so obscured.

 In this manuscript is written about Fusarium wilt of tomato, that caused by Fusarium oxusporum f. sp. lycopersici. Then this pathogen has some races.

 But you wrote only “The pathogen was a Fusarioum oxysporum (F.o) strain kindly provided by INIFAP.” It’s so incomplete explanation.

 Then you used tomato variety only RIO GRANDE. Why did you select this variety?

(Page 10 Line 328-330)

P.10 L. 369: What is INIFAP-Bajío by the Walkeley-Black method? You should show the reference.

    L. 370: I cannot find the reference of Figueroa-Rivera et al. (2010) in “References” in your manuscript.

P.11 L. 380: You showed Table 1, but why eDNA’s levels were only 50 and 500 ppm?

 In bioassay, it’s very important what kind of soil did you use? Is it enough to assess the effect of self-eDNA of fungus in its survival in soil?

P.11 L.415: You wrote that Disease index assay s were classified according to the scale of Bosland and Lindsey. But their scale was for Phytophthora of pepper wilt, not for Fusarium vascular wilt of tomato.

Therefore you should change the assay method for Fusarium wilt of tomato.

Reference [35] is not suitable for this manuscript.

Symptoms of Phytophthora wilt of pepper is different with Fusarium vascular wilt of tomato.

P.11 L.455 – 464: I cannot understand that you use some primers for PCR. But you showed the references only actin primer.

P. 12 L. 426: You wrote about analyses of secondary metabolites.

Then you showed the reference [52], but it was about common bean, not tomato.

As you know, if plants were different, analysis method would be changed often.

Therefore “Results” are not good.    

Especially Figure 4 in P.5 L.163 is not good. Because you showed vascular wilt of tomato, but you use the disease index of phytophthora rot of pepper.

Then you should show the disease index of vascular discoloration.

Figure 5 is also unclear that I cannot see vascular discoloration.

In conclusion, regrettably you should resubmit this manuscript after rewriting.

Best regards,

Comments on the Quality of English Language

This manuscript is necessary to edit minor.

For example, P. 11 L. 412 vernier → vernier caliper

P.12 L. 445 mix → mixture 

Reviewer 2 Report

Comments and Suggestions for Authors

The authors declaimed that application of eDNA of F. o (500 ppm) in soil significant lower disease severity levels in tomato, increasing the content of some phenylpropanoids, and positively regulating the expression of defense genes. However, there are many issues including formatting, some data processing and translation problems needed to be solved in this manuscript.

1. The format all images need to be unified and standardized with full line of Y axis, the authors should double check the whole manuscript.

2. In Figure 1 and 2, “Different letters indicate significant differences (two-way ANOVA and Tukey´s test with α=0.05)” should be “Different lowercase letters indicate significant differences (two-way ANOVA and Tukey´s test with α=0.05)”. In general, different lowercase letters indicate significant differences at 0.05 level, while different uppercase letters indicate significant differences at 0.01 level. The author should change the uppercase letters into lowercase letters to indicate significant differences at 0.05 level in the panels, and might change the lowercase letters, a), b), c), d), for instance, into A, B, C, D to indicate different panels. In addition, no error bars in Fig2b, they should be added.

3. In the legend of Figure 2, CTS-50 from the legend is not found in the panels, instead of CTS-500; while no CTS-2d from the panel is not indicated in the legend. In addition, in control should also indicated in detail in the legend. The authors should double check the whole manuscript.

4. In line 137, “2.2. Vascular wilt evaluation in tomato treated with eDNA of F. oxysporum and S. lycopersicum”, where is the eDNA treatment of S. lycopersicum? So as in 175, “2.3. Biochemical and molecular changes in tomato treated with eDNA of F. oxysporum and S. lycopersicum”, there is no S. lycopersicum treatment.

5. In Figure3, the title of Y axis should be labeled in detail. In Figure4, “Different letters in each group indicate significant differences (one-way ANOVA and Wilcoxon test with α=0.05)”, which is not consistent to the section 4.8. Statistical analysis, “…… tested by a Kruskal-Wallis test and Tukey´s range test in R version 4.1.3”. No Kruskal-Wallis test is found in the context.

6. In Figure 5, what are the differences between the left pictures and the right ones (daker background)? In addition, the pictures seem to be casually taken and put together, with or without white label, or different background, and no scale bars.

7. In Figure 6, “Total flavonoids determination in plants treated with tomato eDNA (a) and F. o eDNA (b)”, but in the panel, only one of them, quercetin, was shown, what about the others? In addition, in upper panel a), “queretin” was shown which is different from quercetin in the lower panel b). Is it typo?

8. So as in Figure 7, gallic acid is no the total phenols, but one of them. The authors should also check the context. In addition, “F. o eDNA (b)” should be “F. o eDNA (b)”. AndPT= preventive control, CT= corrective control” should be “PT= preventive treatment, CT= corrective treatment”.

9. In Figure8, why the author only detected “tomato eDNA = preventive treatment with tomato eDNA, and F. o eDNA = preventive treatment with F. o eDNA in tomato”, without detecting the “corrective treatment of tomato eDNA, and F. o eDNA”?  

10. There are lots errors in the context, for instance: extra “or” in line 20 “……danger by or damage-associated molecular patterns (DAMP)”;

In Line370, citation format error;

In line 69, “Ca2+ cations”;

In line 146, “(data not shown) (Figure 5d)”, the description is confused;

In line 149, “in figure 5 is shown……” should be “In figure 5 is shown……”

In line 371, “2x109 spores/ml of F. o” should be “2x109 spores/mL of F. o

In Line461, "15seg" should be "15s".

Please double check and correct all the errors.

Comments on the Quality of English Language

The authors declaimed that application of eDNA of F. o (500 ppm) in soil significant lower disease severity levels in tomato, increasing the content of some phenylpropanoids, and positively regulating the expression of defense genes. However, there are many issues including formatting, some data processing and translation problems needed to be solved in this manuscript.

1. The format all images need to be unified and standardized with full line of Y axis, the authors should double check the whole manuscript.

2. In Figure 1 and 2, “Different letters indicate significant differences (two-way ANOVA and Tukey´s test with α=0.05)” should be “Different lowercase letters indicate significant differences (two-way ANOVA and Tukey´s test with α=0.05)”. In general, different lowercase letters indicate significant differences at 0.05 level, while different uppercase letters indicate significant differences at 0.01 level. The author should change the uppercase letters into lowercase letters to indicate significant differences at 0.05 level in the panels, and might change the lowercase letters, a), b), c), d), for instance, into A, B, C, D to indicate different panels. In addition, no error bars in Fig2b, they should be added.

3. In the legend of Figure 2, CTS-50 from the legend is not found in the panels, instead of CTS-500; while no CTS-2d from the panel is not indicated in the legend. In addition, in control should also indicated in detail in the legend. The authors should double check the whole manuscript.

4. In line 137, “2.2. Vascular wilt evaluation in tomato treated with eDNA of F. oxysporum and S. lycopersicum”, where is the eDNA treatment of S. lycopersicum? So as in 175, “2.3. Biochemical and molecular changes in tomato treated with eDNA of F. oxysporum and S. lycopersicum”, there is no S. lycopersicum treatment.

5. In Figure3, the title of Y axis should be labeled in detail. In Figure4, “Different letters in each group indicate significant differences (one-way ANOVA and Wilcoxon test with α=0.05)”, which is not consistent to the section 4.8. Statistical analysis, “…… tested by a Kruskal-Wallis test and Tukey´s range test in R version 4.1.3”. No Kruskal-Wallis test is found in the context.

6. In Figure 5, what are the differences between the left pictures and the right ones (daker background)? In addition, the pictures seem to be casually taken and put together, with or without white label, or different background, and no scale bars.

7. In Figure 6, “Total flavonoids determination in plants treated with tomato eDNA (a) and F. o eDNA (b)”, but in the panel, only one of them, quercetin, was shown, what about the others? In addition, in upper panel a), “queretin” was shown which is different from quercetin in the lower panel b). Is it typo?

8. So as in Figure 7, gallic acid is no the total phenols, but one of them. The authors should also check the context. In addition, “F. o eDNA (b)” should be “F. o eDNA (b)”. AndPT= preventive control, CT= corrective control” should be “PT= preventive treatment, CT= corrective treatment”.

9. In Figure8, why the author only detected “tomato eDNA = preventive treatment with tomato eDNA, and F. o eDNA = preventive treatment with F. o eDNA in tomato”, without detecting the “corrective treatment of tomato eDNA, and F. o eDNA”?  

10. There are lots errors in the context, for instance: extra “or” in line 20 “……danger by or damage-associated molecular patterns (DAMP)”;

In Line370, citation format error;

In line 69, “Ca2+ cations”;

In line 146, “(data not shown) (Figure 5d)”, the description is confused;

In line 149, “in figure 5 is shown……” should be “In figure 5 is shown……”

In line 371, “2x109 spores/ml of F. o” should be “2x109 spores/mL of F. o

In Line461, "15seg" should be "15s".

Please double check and correct all the errors.

Reviewer 3 Report

Comments and Suggestions for Authors

In this manuscript, the authors investigated the role of the extracellular DNA (eDNA) in inhibiton to Fusarium oxysporum and in induction of tomato defense. However, the manuscript has serious flaws, it cannot be accepted for publishing at current version.

The major concerns:

1) The tomato seedlings in Figure 5 do not grow well, look like have stress even in the control, which seriously effect the results.

2) Some results are unconvincing or misinterpretation:

Line 102-105: the explanation is opposite with the Figure 1 result. In the Figure, the PTS-500 show higher growth.

Line 146: Is it wrong description? Figure 5d is wounded positive control, no eDNA was added.

Line 148: It is a wrong description. Becasue the PT and CT treatments show less severity level than wounded positive control in Figure 4a.

Line 149-153: from the Figure 5, I cannot see the phenotypes decribed by authors, new Figure need.

Line 182-184: It is a wrong description. The wounded control also increases the flavonoids level in the Figure 6a, but the CT treatment is the same level as control in Figure 6b.

But if comparing with WPC, the flavonoids levels in both CT and PT treatment are almost same, how can the authors get the conclusion that eDNA  can induce flavonoids levels?

Line 195-197:  In the Figure 7a,  the WC treatment show highest phenols level, can it still get this conclusion?

Line 197-201: In the Figure 7b, the WPC and CT show the same  phenols level, how to get this conclusion?

Line 218-219: It is a wrong description. In the Figure 8a, the gene expression level is too low in F. o eDNA treatment and even lower than WPC.  Same as in line 221-224, cannot get this conclusion.

Line 281-286, 299-300: the conclusions are wrong just as the above.

In the Materials and Methods section: 

1) How to evaluate the fungal spores and vegetative tissue viability in soil?

2) In the Table 2, Corrective treatment, why use 6 ml eDNA but no 2 ml?

Reviewer 4 Report

Comments and Suggestions for Authors

see attached file.

Comments on the Quality of English Language

needs to be improved.